# Creating and Leveraging a Synthetic Dataset of Cloud Optical Thickness Measures for Cloud Detection in MSI

Aleksis Pirinen [1,*], Nosheen Abid [2], Nuria Agues Paszkowsky [1], Thomas Ohlson Timoudas [1], Ronald Scheirer [3], Chiara Ceccobello [4], György Kovács [2] and Anders Persson [5]

1. Department of Computer Science, RISE Research Institutes of Sweden, 501 15 Borås, Sweden; nuria.agues.paszkowsky@ri.se (N.A.P.); thomas.ohlson.timoudas@ri.se (T.O.T.)
2. Machine Learning, Department of Computer Science, Electrical and Space Engineering, Luleå University of Technology, 971 87 Luleå, Sweden; nosheen.abid@ltu.se (N.A.); gyorgy.kovacs@ltu.se (G.K.)
3. Swedish Meteorological and Hydrological Institute, 601 76 Norrköping, Sweden; ronald.scheirer@smhi.se
4. AI Sweden, 417 56 Gothenburg, Sweden; chiara.ceccobello@ai.se
5. The Swedish Forest Agency, 551 83 Jönköping, Sweden; anders.persson@skogsstyrelsen.se
* Correspondence: aleksis.pirinen@ri.se

**Abstract:** Cloud formations often obscure optical satellite-based monitoring of the Earth's surface, thus limiting Earth observation (EO) activities such as land cover mapping, ocean color analysis, and cropland monitoring. The integration of machine learning (ML) methods within the remote sensing domain has significantly improved performance for a wide range of EO tasks, including cloud detection and filtering, but there is still much room for improvement. A key bottleneck is that ML methods typically depend on large amounts of annotated data for training, which are often difficult to come by in EO contexts. This is especially true when it comes to cloud optical thickness (COT) estimation. A reliable estimation of COT enables more fine-grained and application-dependent control compared to using pre-specified cloud categories, as is common practice. To alleviate the COT data scarcity problem, in this work, we propose a novel synthetic dataset for COT estimation, which we subsequently leverage for obtaining reliable and versatile cloud masks on real data. In our dataset, top-of-atmosphere radiances have been simulated for 12 of the spectral bands of the Multispectral Imagery (MSI) sensor onboard Sentinel-2 platforms. These data points have been simulated under consideration of different cloud types, COTs, and ground surface and atmospheric profiles. Extensive experimentation of training several ML models to predict COT from the measured reflectivity of the spectral bands demonstrates the usefulness of our proposed dataset. In particular, by thresholding COT estimates from our ML models, we show on two satellite image datasets (one that is publicly available, and one which we have collected and annotated) that reliable cloud masks can be obtained. The synthetic data, the newly collected real dataset, code and models have been made publicly available.

**Keywords:** cloud detection; machine learning; cloud optical thickness; datasets

## 1. Introduction

Space-borne Earth observation (EO) has vastly improved our options for collecting information from the world we live in. Not only atmospheric parameter from remote places, but also layers underneath the atmosphere are subject to these remote sensing applications. For example, land use and land cover classification [1], damage assessment for natural disasters [2,3], biophysical parameter-retrieval [4], urban growth monitoring [5], and crop yield estimation [6] are globally derived on a regular basis. Most of these applications rely on optical sensors of satellites gathering images, and the cloud coverage is a barrier that hampers the exploitation of the measured signal [7]. Accurate cloud cover estimation is essential in these applications, as clouds can critically compromise their performance.

The cloud detection and estimation solutions in multispectral images range from rule-based statistical methods to advanced deep learning approaches. On the one hand, rule-based thresholding methods exploit the physical properties of clouds reflected on different spectral bands of satellite images to generate cloud masks. FMask [8] and Sen2Cor [9] are examples of such approaches implemented for cloud masking for Landsat and Sentinel-2 multispectral imagery, respectively. On the other hand, recently, there has been a burst of work in the literature that uses ML approaches to solve the cloud detection and estimation problem [10–13]. These works use convolutional neural networks with large manually annotated datasets of satellite images and perform better than statistical rule-based methods.

Clouds are inhomogeneous by nature and the spatial inhomogeneity of clouds, or their varying cloud optical thickness (COT), affects not only the remote sensing imagery but also atmospheric radiation. The COT is an excellent proxy for a versatile cloud mask. By choosing a threshold value for the cloud/clear distinction (or even finer-grained categories), both a cloud-conservative and a clear-conservative cloud mask can be implemented. Furthermore, a continuous target has advantages over a discrete one when training using machine learning (ML). Most literature estimates COT using independent pixel analysis (IPA, also known as ICA, independent column approximation [14]). IPA considers the cloud properties homogeneous in the pixel and carries no information about the neighboring pixels. Statistical approaches [15–17] have explored the potential causes influencing COT and defined parameters to mitigate their effect. With the advancement of deep learning [18,19], some approaches for COT using neural networks have been proposed [20,21]. Most of these methods require neighboring information of the pixels and large amounts of annotated spatial data.

Both statistical and ML-based methods commonly require benchmark datasets to evaluate their performance and find areas of improvement. Each pixel in the satellite images must be labeled manually or by a superior instrument (e.g., an active LIDAR [22]) as cloudy or cloud-free (or using even finer-grained labels). The complex nature of clouds makes the task even more difficult. This type of labeling is generally time-consuming and requires expert domain knowledge, leading to limited publicly available datasets.

*Contributions*

Below, we provide details of the contributions of this work, which can be summarized as (i) the creation and public release of a synthetic 1D dataset simulating top-of-atmosphere (TOA) radiances for 12 of the spectral bands of the Multispectral Imagery (MSI) onboard the Sentinel-2A platform; and (ii) an extensive set of experiments with different ML approaches, which show that COT estimation can be used as a flexible proxy for obtaining reliable and versatile cloud masks on real data.

As for (i), we propose a new 1D dataset simulating TOA radiances for 12 of the spectral bands (the aerosol (B1) band is not included in the simulation, as it adds another dimension of computational complexity to the simulation; its inclusion is an avenue for future work) of the MSI onboard the Sentinel-2A platform. The data points have been simulated, taking into consideration different cloud types, cloud optical thicknesses (COTs), cloud geometrical thickness, cloud heights, water vapor contents, gaseous optical thicknesses, as well as ground surface and atmospheric profiles. To this end, we connected the resources natively available in RTTOV v13 [23] with external resources, such as a dataset of atmospheric profiles provided by ECMWF, and a dataset of spectral reflectances from the ECOSTRESS spectral library [24,25]. More details about the dataset are provided in Section 2.

The public release of these data implies that the threshold of research participation is lowered, in particular for non-domain experts who wish to contribute to this field, and offers reproducible and controllable benchmarking of various methods and approaches. Potential users of our dataset should be aware of the possible limitations the IPA could introduce in their study. In that regard, we note that if the users are interested in developing

high-spatial-resolution applications, IPA will introduce a systematic error in COT, due to 3D effects [26]. Nevertheless, the derivation of COT allows for more flexible use of data (e.g., clear-conservative vs. cloud-conservative cloud mask).

As for (ii), we present results for a range of ML models trained for cloud detection and COT estimation using our proposed dataset. The experimental results suggest that on unseen data points, multi-layer perceptrons (MLPs) significantly outperform alternative approaches like linear regression. We have also investigated and demonstrated the performance of our trained models on two real satellite image datasets—the publicly available KappaZeta, ref. [27] as well as a novel dataset provided by the Swedish Forest Agency (which has also been made publicly available). These results show that by thresholding COT estimates from our ML models trained on the proposed synthetic dataset, reliable cloud masks can be obtained on real images; they even outperform the scene classification layer by ESA on the dataset provided by the Swedish Forest Agency. The main ML models are described in Section 3 and the experimental results are included in Section 4.

## 2. Synthetic Cloud Optical Thickness Dataset

In this section, we explain the COT synthetic dataset that we propose to use for subsequent threshold-based cloud masking. The dataset consists of 200,000 simulated data points (akin to individual and independent pixels, as observed by a satellite instrument), which have been simulated taking into consideration different cloud types, cloud optical thicknesses (COTs), cloud geometrical thickness, cloud heights, as well as ground surface and atmospheric profiles. Variations in water vapor content, gaseous optical thickness, and angles (azimuth difference as well as satellite and solar zenith angle) have also been included in the dataset.

To generate these data, a connection had to be established between surface and atmospheric properties on the one hand, and on the other hand, top-of-atmosphere (ToA) reflectivities, as observed by the MSI at Sentinel-2A. We achieved this using a radiative transfer model, namely the fast Radiative Transfer for TOvs. (RTTOV) v13 [23]. The simulations are performed for the instrument on the first of a series of Sentinel-2 platforms. Due to variations in between different realizations of the MSI and because of degradation of the instrument over time, it is recommended to add some noise to the reflection data before training to obtain more robust models (see also Sections 3.1 and 4).

The backbone of each simulation is an atmospheric profile defined by the vertical distribution of pressure, temperature, water vapor content and ozone. These atmospheric profiles are picked at random from an ECMWF (European Centre for Medium-range Weather Forecasts) dataset of 10,000 profiles compiled with the aim of mapping the widest possible range of atmospheric conditions [28]. The surface is assumed to work as a lambertian reflector [29]. All spectral reflectance is provided by the ECOSTRESS spectral library [24,25]. The sun and satellite angles are varied randomly.

Each main surface type is composed of several sub-groups. The main surface types are *non-photosynthetic vegetation* (e.g., tree bark, branches, flowers), *rock* (e.g., gneiss, marble, limestone), *vegetation* (e.g., grass, shrub, tree), *soil* (e.g., calciorthid, quartzipsamment, cryoboroll) and *water* (ice, snow in different granularities, and liquid). The total number of different reflectivity profiles (including all not-listed sub-groups) comprises 743 variations. These 743 fine classes appear in the dataset as 139 distinguishable categories. The distribution of the main types within the final dataset is shown in Table 1. The instrument-specific reflection was derived by convolving the spectral surface reflection with the spectral response function of the corresponding MSI channel. When generating the data, surface types were sampled at random.

**Table 1.** Distribution of the main surface types within our dataset. Details and examples of subgroups are given in the main text.

| Main Surface Type | Frequency |
| --- | --- |
| Vegetation | 70.5% (140,984) |
| Rock | 10.7% (21,319) |
| Non-photosynthetic veg. | 7.9% (15,790) |
| Water | 5.8% (11,526) |
| Soil | 5.2% (10,381) |

The dataset consists of four equal-sized parts (50,000 points each). One part features clear sky calculations only (no clouds are considered here, only atmospheric gases and surface properties), while the three remaining parts respectively depict water clouds, ice clouds, and mixed clouds. The cloud geometrical thickness is set at random. The clouds' geometrical thickness varies from one layer to six with the mode at two and three. Clouds of the same type populate adjacent layers, but it is possible to have vertical gaps between ice and water clouds. The cloud type (for water clouds) and liquid water/ice content is varied randomly. Water clouds are parameterized following the size distributions of the five OPAC cloud types [30] with updates in the liquid water refractive index dataset according to [31]. Ice clouds are parameterized by temperature and ice water content (IWC) according to [32]. Variations in size and shape distributions are taken into account implicitly by the parametrization of particle optical properties. Distributions of COT for water, ice, and combined clouds are shown in Figure 1, where it can be seen that our focus is set on optically thin clouds. Note that when a consistency check fails (i.e., the cloud type and background atmospheric temperature do not match), the cloud will be removed and the COT is set to zero, but the data point remains in its place. Thus, within the dataset there are some data points that are labeled as cloudy, even though they depict clear sky cases. These clear sky cases are removed from the plots in Figure 1.

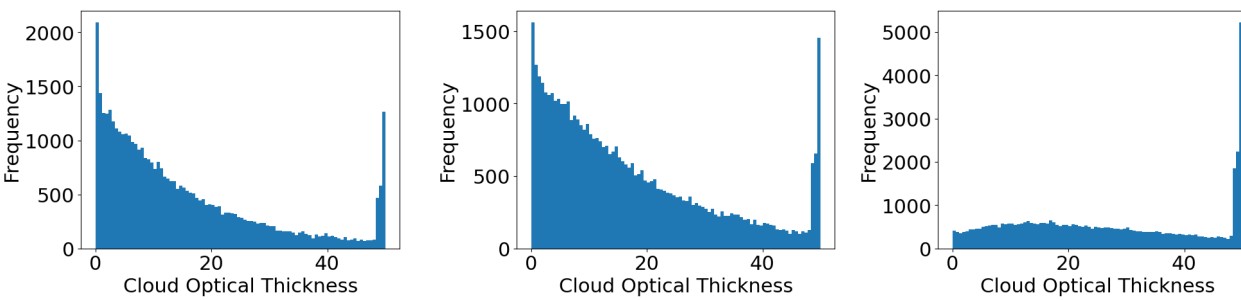

**Figure 1.** Distribution of COTs in the range $[0, 50]$ for different cloud types in our dataset: water clouds (**left**), ice clouds (**center**), and combined water and ice clouds (**right**). The focus of our data is on optically thin clouds.

The overall cap of COT $\leq 50$ is a feature of the radiative transfer model. This is not seen as a limitation for model training, because COT is mainly used as a means for subsequent cloud classification. For example, see Section 4.2, where we successfully perform classification of real image pixels into semi-transparent and opaque (optically thin and thick) clouds. We do this even though we understand that the pure prediction of the optical density, due to 3D effects, has little significance. The most relevant 3D effect for MSI with a high spatial resolution is the neglect of the net horizontal radiation transport. The error depends on the horizontal component of the photon movement, and thus largely on the solar zenith angle. However, even the incorrect COT is suitable for a cloudy/cloud-free decision. Note that according to the international satellite cloud climatology project (ISCCP, see e.g., [33]), clouds are considered optically thin for COT $< 3.6$, medium for $3.6 \leq \text{COT} < 23$, and optically thick for COT $\geq 23$. Keeping in mind that optically thick

clouds are in general easier to detect, it is not expected that this limitation will affect our conclusions.

In summary, our dataset $\mathcal{D} = \{(x_1, y_1), \ldots, (x_n, y_n)\}$ consists of 200,000 pairs $(x_i, y_i)$ with $x_i \in \mathbb{R}^{19}$ and $y_i \in \mathbb{R}_+$ the $i$:th input and ground truth COT, respectively, i.e., each element in the 19-dimensional vector $x_i$ is a real scalar, and $y_i$ is a non-negative real scalar (since COT cannot be negative). By ground truth COT $y_i$, we mean the true COT against which COT predictions can be compared (e.g., measure how accurately a machine learning model predicts $y_i$ if it receives $x_i$ as input). As for $x_i$, it contains simulated measured reflectivities for the 12 spectral bands, but also additional optional features: (i) satellite zenith angle [°]; (ii) sun zenith angle [°]; (iii) azimuth difference angle [°]; (iv) gas optical thickness [−]; (v) vertically integrated water vapor [g/cm$^2$]; (vi) surface profile [−]; and (vii) cloud type [−].

In this work, we have focused on experiments using only the 12 band reflectivities as model input, which are least effected by aerosols, and thus from this point onwards we will let $x_i \in \mathbb{R}^{12}$ denote such a data point. One of the main reasons for not using other types of input information besides the band reflectivities is compatibility (to ensure that the resulting models would be able to be evaluated on publicly available real image datasets, see, e.g., Section 4.2). As for the ground truth COT $y_i$, these lie in the range $y_i \in [0, 50]$; again, the upper limit is set by the radiative transfer model. We randomly set aside 160,000 data points for model training, 20,000 for validation, and 20,000 for testing (see Sections 3.1 and 4).

## 3. Machine Learning Models

In this section, we briefly introduce the different machine learning (ML) models that we have implemented for the COT estimation task. Recall from Section 2 that the model inputs $x_i \in \mathbb{R}^{12}$ contain only 12 band reflectivities, i.e., no auxiliary inputs such as surface profile information are used by the models. As our proposed dataset can be seen as consisting of individual 'pixels', with no spatial relationship in between them (cf. Section 2), we have mainly considered multi-layer perceptron (MLP) models (early experiments were performed using a range of approaches, e.g., random forests, but MLPs performed best). Extensive model validations suggested that a five-layer MLP with ReLU-activations and hidden dimension 64 (the same for all layers) yields the best results. See Figure 2 for an overview of this architecture.

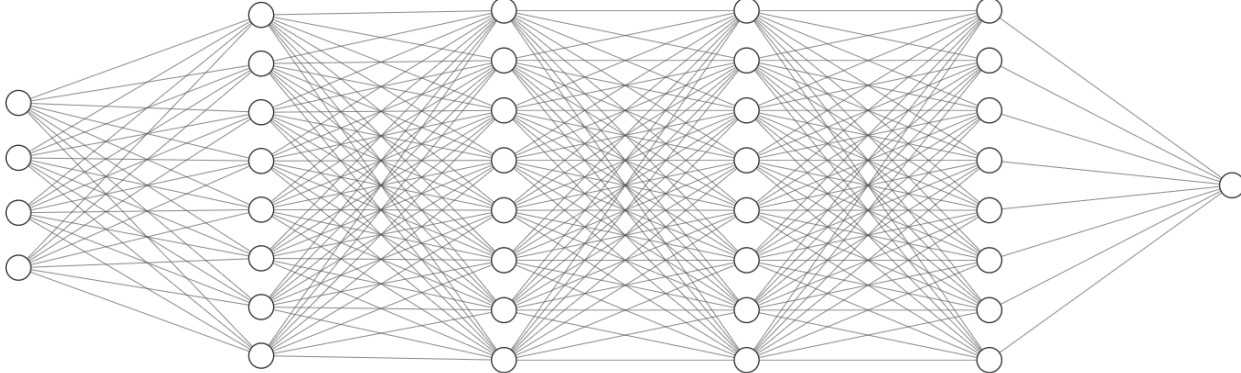

**Figure 2.** Main MLP model architecture that we use for COT estimation. The input layer is shown on the left (4 nodes are shown for the input $x$ instead of all 12, to avoid visual clutter). This is followed by four hidden layers (8 nodes shown for each, instead of 64). Finally, the output layer on the right produces the COT estimate $\hat{y}$. Each of the five layers is of the form $z^k = \text{ReLU}(W^k z^{k-1} + b^k) = \max(0, W^k z^{k-1} + b^k)$ for $k = 1, \ldots, 5$, with $z^0 = x$ and $z^5 = \hat{y}$. Here, $W^k$ and $b^k$ respectively represent the weight matrix and bias vector for the $k$:th layer, and these parameters are calibrated through the training process (see Section 3.1). Note that we use an MLP, rather than, e.g., a convolutional architecture, since the data points $x_i$ are independent from each other in our synthetic dataset.

When performing inference on real imagery $I \in \mathbb{R}^{H \times W \times C}$ with an MLP model, we let it operate at a pixel-per-pixel basis, as the models are trained (see Section 3.1) on *individual* synthetic pixels due to the nature of the synthetic data. To improve spatial consistency of a resulting COT prediction map $C \in \mathbb{R}^{H \times W}$, we apply a simple post-processing scheme where a sliding window of size $M \times M$ and stride 1 runs on top of $C \in \mathbb{R}^{H \times W}$, and produces an average (among $M^2$ values) at each location. Overlaps that occur due to the stride being smaller than the width of the kernel are handled by averaging. We found $M = 2$ to produce robust results. Thus, during this post-processing step, each value within $C$ is essentially replaced by the mean value of the surrounding values, which increases the smoothness of the predictions (nearby predictions are less likely to differ drastically). This is desirable, because in real images, adjacent pixels are typically quite similar to each other, so the predictions for adjacent pixels are expected to be quite similar as well.

### 3.1. Model Training

As is standard within machine learning, we normalized the model inputs $x_i \in \mathbb{R}^{12}$ such that each of the 12 entries of $x_i$ had zero mean and unit standard deviation. This was achieved by first computing the mean $x^{\text{mean}} \in \mathbb{R}^{12}$ and standard deviation $x^{\text{std}} \in \mathbb{R}^{12}$ on the training set, and then for each $x_i$, applying the following transformation (the subtraction and division below were performed element-wise for each of the 12 entries) before feeding the result to the model:

$$x_i \leftarrow \frac{x_i - x^{\text{mean}}}{x^{\text{std}}}. \tag{1}$$

We also explored other data normalization techniques but did not see any improvements relative to (1). To improve model robustness, independent Gaussian noise was also added to each input. This noise was zero-mean and had a standard deviation corresponding to 3% of the average magnitude of each input feature. Thus, the transformation (1) was replaced by

$$x_i \leftarrow \frac{(x_i + \varepsilon_i) - x^{\text{mean}}}{x^{\text{std}}}, \tag{2}$$

where $\varepsilon_i \in \mathbb{R}^{12}$ denotes such an independently sampled Gaussian.

The ML models are trained with the commonly used Adam optimizer [34] with a mean squared error (MSE) loss for 2,000,000 batch updates (batch size 32, learning rate 0.0003). Since the models are very lightweight, training can be carried out even without a graphics processing unit (GPU) within about an hour.

### 3.2. Fine-Tuning with Weaker Labels

For some datasets (see Section 4.2), we have access to weaker labels in the form of pixel-level cloud masks. For instance, in the KappaZeta dataset [27], there are labels for 'clear', 'opaque cloud', and 'semi-transparent cloud', respectively. In addition to evaluating models on such data, we can also set aside a subset of those data to perform model refinement based on the weaker labels.

Let $\tau^{\text{semi}}$ and $\tau^{\text{opaque}}$ (where $0 < \tau^{\text{semi}} < \tau^{\text{opaque}}$) denote the semi-transparent and opaque COT thresholds, respectively. Then, we refine a model using a loss $\mathcal{L}$ which satisfies the following criteria. If $p$ denotes the prediction of a pixel which is labeled as

- 'clear', then

$$\mathcal{L}(p) = \begin{cases} 0 & \text{if } p \leq \tau^{\text{semi}}, \\ \frac{1}{2}(p - \tau^{\text{semi}})^2 & \text{if } p > \tau^{\text{semi}}; \end{cases}$$

- 'opaque cloud', then

$$\mathcal{L}(p) = \begin{cases} 0 & \text{if } p \geq \tau^{\text{opaque}}, \\ \frac{1}{2}(p - \tau^{\text{opaque}})^2 & \text{if } p < \tau^{\text{opaque}}; \end{cases}$$

- 'semi-transparent cloud', then

$$\mathcal{L}(p) = \begin{cases} 0 & \text{if } \tau^{\text{semi}} \le p \le \tau^{\text{opaque}}, \\ \frac{1}{2}(p - \tau^{\text{semi}})^2 & \text{if } p < \tau^{\text{semi}}, \\ \frac{1}{2}(p - \tau^{\text{opaque}})^2 & \text{if } p > \tau^{\text{opaque}}. \end{cases}$$

## 4. Experimental Results

In this section, we extensively evaluate various ML models discussed in Section 3. For COT estimation, we evaluate the following models:

- *MLP-k* is a *k*-layer MLP (the main model has $k = 5$, cf. Figure 2), where the results are averaged among 10 identical *k*-layer MLP (only differing in the random seed for the network initialization);
- *MLP-k-ens-n* is an ensemble of *n* *k*-layer MLPs, each identically trained but with a unique random network initialization;
- *MLP-k-no-noise* is the same as *MLP-k*, but is trained without adding any noise to the training data (recall that we train with 3% additive noise by default);
- *lin-reg* is a linear regression model.

### 4.1. Results on Synthetic Data

For our proposed synthetic dataset (cf. Section 2), we have access to ground truth COTs, and thus in Table 2, we report the root mean square error (RMSE) between the predicted outputs and corresponding ground truths on unseen test data. We see that training on data with artificially added noise improves model robustness significantly, whereas ensembling only marginally improves performance. Also, MLPs significantly outperform linear regression models.

**Table 2.** RMSE values on different variants of the test set of our synthetic dataset. *Test-x%* refers to the test set with *x*% added noise. Ensemble methods marginally improve over single-model ones. Models trained with additive input noise yield significantly better average results. Linear regression performs worst by far. Note that for the single-model variants MLP-5, MLP-5-no-noise and Lin-reg, we show the mean RMSE over 10 different network parameter initializations and standard deviations.

| Dataset | MLP-5 | MLP-5-ens-10 | MLP-5-no-noise | MLP-5-no-noise-ens-10 | Lin-reg |
|---|---|---|---|---|---|
| Test-0% | $1.63 \pm 0.01$ | 1.56 | $1.05 \pm 0.01$ | 0.92 | $6.49 \pm 0.00$ |
| Test-1% | $1.68 \pm 0.01$ | 1.61 | $1.59 \pm 0.01$ | 1.46 | $6.51 \pm 0.00$ |
| Test-2% | $1.82 \pm 0.01$ | 1.75 | $2.51 \pm 0.01$ | 2.34 | $6.55 \pm 0.00$ |
| Test-3% | $2.04 \pm 0.01$ | 1.97 | $3.42 \pm 0.02$ | 3.20 | $6.63 \pm 0.00$ |
| Test-4% | $2.32 \pm 0.01$ | 2.25 | $4.21 \pm 0.03$ | 3.95 | $6.74 \pm 0.01$ |
| Test-5% | $2.63 \pm 0.01$ | 2.56 | $4.90 \pm 0.04$ | 4.58 | $6.88 \pm 0.01$ |
| Average | $2.02 \pm 0.01$ | 1.95 | $2.95 \pm 0.02$ | 2.74 | $6.63 \pm 0.00$ |

As mentioned in Section 2, our synthetic dataset covers five main surface profiles: vegetation, rock, non-photosynthetic vegetation, water, and soil. Also recall that vegetation is by far the most common main profile in the dataset (70.5% of the data), whereas rock is the second most common one (10.7% of the data). To investigate how the prediction accuracy of the best model MLP-5-ens-10 performs on these five different main profiles, we also compute five main profile-conditioned test results (averaged across the different noise percentages, cf. bottom row of Table 2). We find that the RMSEs were 1.45, 3.97, 2.64, 3.44, and 2.42 for vegetation, rock, non-photosynthetic vegetation, water and soil, respectively. Thus, the model performs best on vegetation data points and worst on data points that depict rock.

*4.2. Results on Real KappaZeta Data*

This publicly available dataset of real satellite imagery [27] is used to assess the real-data performance of models that were trained on our synthetic dataset. Since ground truth information about COT is not available, we consider the weaker pixel-level categories 'clear', 'semi-transparent cloud' and 'opaque cloud'. We set aside a subset (April, May, June; 3543 images that we refer to as the training set) in which we search for COT thresholds which correspond to semi-transparent and opaque cloud, respectively. These thresholds are respectively found to be 0.75 and 1.25 (both values in the range $[0, 50]$, cf. Section 2).

Given these thresholds, we evaluate a model's semantic segmentation performance (into the three aforementioned cloud categories) during July, August and September (2511 images in total, which we refer to as the test set; there is no spatial overlap between the training and evaluation sets). We also conduct similar experiments in which we first refine the MLP models on the training set of KappaZeta, cf. Section 3.2. Such a model is denoted as *MLP-k-tune*. This comparison allows us to quantify how much is gained in prediction accuracy if one trains on the target data domain (the training set of KappaZeta), as opposed to training only on off-domain data (our synthetic dataset). We also compare our MLP models with a U-net segmentation network, for which we use the open-source *FCN8* model [35] for the same task. Unlike an MLP, a U-net is a fully convolutional architecture that explicitly incorporates contextual information from surrounding pixels in an image when performing predictions. Thus, this comparison with a U-net allows us to quantify how much is gained in prediction accuracy if one uses a model that takes into account spatial connectivity among pixels—we expect the U-net to be superior, since U-nets incorporate and take advantage of spatial patterns in images, while the MLPs inherently cannot.

The results are shown in Table 3, where two metrics are reported for each cloud category:

- F1-score, i.e. the harmonic mean of precision and recall, where precision is the fraction of pixels that were correctly predicted as the given category, and recall is the fraction of pixels of the given category that were predicted as that category;
- Intersection over union (IoU), i.e. the ratio between the number of pixels in the intersection and union, respectively, of the predicted and ground truth cloud masks.

**Table 3.** Results on the KappaZeta test set. MLP approaches do not perform as well as U-nets, as is expected, since U-nets integrate information from other pixels, which MLPs inherently cannot. This can also be seen in the qualitative results in Figure 3. Among the MLP approaches, model fine-tuning on the KappaZeta training set only slightly improves the results (column 4 vs. 3), i.e., our synthetic dataset allows for model generalization to real data, while ensembling tuned models negligibly improves results (column four vs. two).

| Metric | MLP-5-tune | MLP-5-ens-10 | MLP-5-ens-10-tune | U-Net |
|---|---|---|---|---|
| F1-avg | 0.51 | 0.49 | 0.52 | 0.66 |
| F1-clear | 0.54 | 0.53 | 0.54 | 0.72 |
| F1-semi | 0.30 | 0.25 | 0.31 | 0.49 |
| F1-opaque | 0.70 | 0.70 | 0.71 | 0.78 |
| mIoU | 0.46 | 0.43 | 0.47 | 0.54 |
| IoU-clear | 0.63 | 0.60 | 0.63 | 0.62 |
| IoU-semi | 0.20 | 0.15 | 0.21 | 0.35 |
| IoU-opaque | 0.54 | 0.54 | 0.56 | 0.65 |

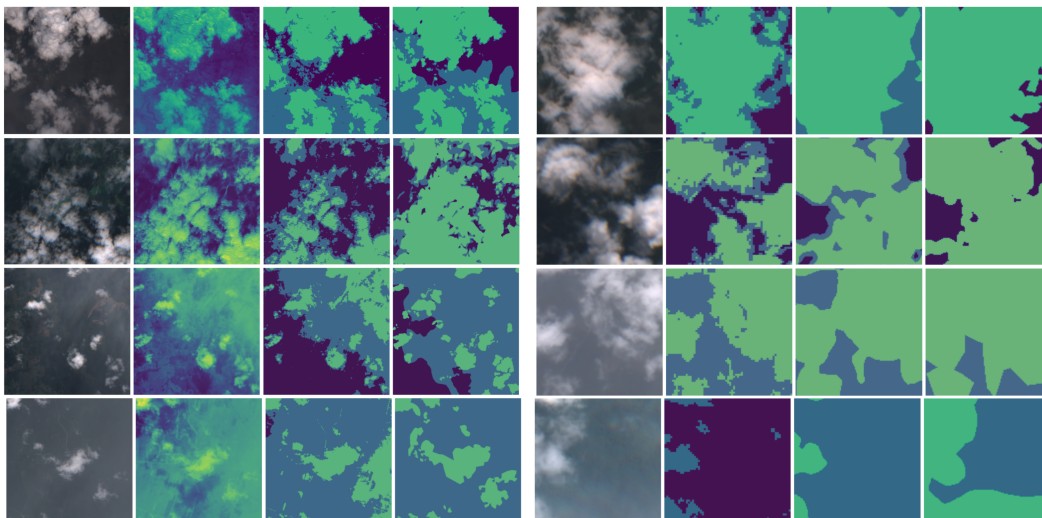

**Figure 3.** (**Left**): Examples of our main MLP approach (ensemble of ten 5-layer MLPs) on unseen KappaZeta test data. Column 1: Input image (only RGB is shown). Column 2: COT estimates (relative intensity scaling, to more clearly show the variations). Column 3: Pixel-level cloud type predictions based on thresholding the COTs in column 2. Column 4: KappaZeta ground truth. Dark blue is clear sky, lighter blue is semi-transparent cloud, and turquoise is opaque cloud. (**Right**): Similar to the left, but the 2nd column shows the thresholded model predictions (instead of the COT estimates), and the 3rd column is the U-net prediction. A failure case for both models is shown on the bottom row.

Both metrics are bounded in the $[0, 1]$-range (higher is better). For both metrics, we also show the average result over the different cloud types (these are respectively denoted as F1-avg and mIoU), where the average is computed uniformly across the different cloud types, i.e., each category contributes equally to the mean, independently of how common the category is in the dataset.

We see that the best MLP-approach (MLP-5-ens-10-tune) obtains a lower mIoU and F1-score compared to the U-net (0.47 vs. 0.54 and 0.52 vs. 0.66, respectively). We re-emphasize, however, that the MLP approaches perform *independent* predictions per pixel, whereas the U-net takes spatial context into consideration. Furthermore, we see that model fine-tuning on the KappaZeta training set only slightly improves results on the test set (column 4 vs. column 3), which shows that our synthetic dataset can be used to train models which generalize well to cloud detection on real data. It can also be seen that model ensembling only marginally improves results for KappaZeta-tuned models (column four vs. column two).

As for model inference speeds, the CPU runtime of an MLP model is 0.05 s per image (spatial extent $128 \times 128$). The corresponding runtime for the U-net is 0.65 s per image—thus note that the *MLP approach is 13 times faster on a CPU*. On the GPU, the corresponding runtimes are 0.004 and 0.003 seconds for the MLP and U-net, respectively.

Qualitative Results

We further examine the results by inspecting qualitative examples in Figure 3. The examples on the left side focus on our MLP approach (the one that was only trained on synthetic data) and include the estimated COT values (column 2). On the right side, we include comparisons to the U-net. From these examples, we see that the ground truth segmentation masks are 'biased' towards spatial connectivity among pixels—in many cases, the ground truth does not seem entirely accurate, and rather places emphasis on spatial consistency at the cost of finer-grained contours. This bias is easily incorporated in an architecture which can leverage spatial connectivity (the U-net in our case), whereas it will cause confusion for per-pixel models (the MLPs in our case). In many of the examples, it appears as if the MLP predictions are more accurate than the ground truth when visually

compared with the associated input images (the main exception is shown in row #4 (right) where the MLP approach underestimates the COTs).

In Figure 4, we show qualitative examples of the effect of model refinement on the KappaZeta training set. As can be seen, in many cases, results do not improve by model fine-tuning, i.e., our synthetic dataset can on its own provide models that perform well on many real examples. Recall that this can also be seen quantitatively by comparing the third and fourth columns of Table 3, where one sees that the KappaZeta-tuned variant obtains only slightly higher F1 and mIoU scores.

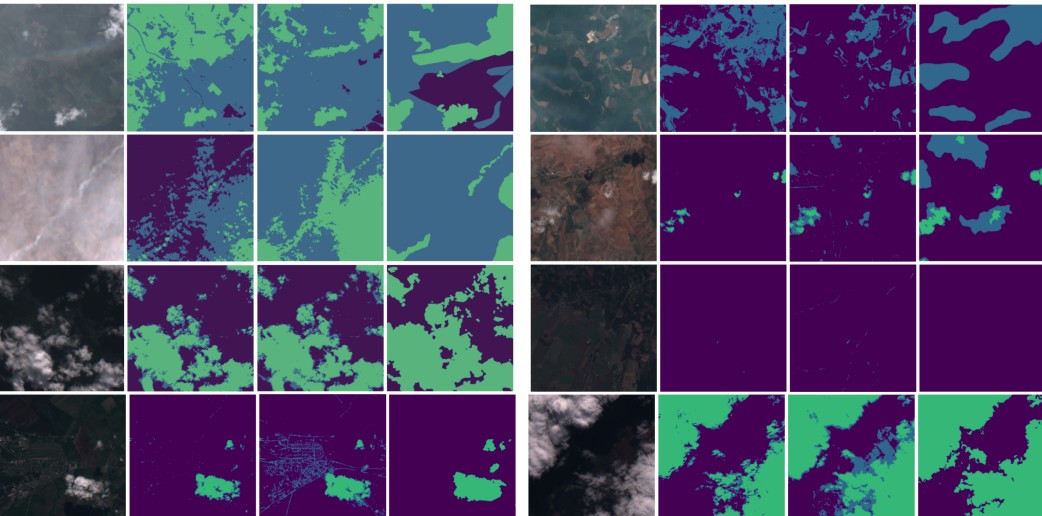

**Figure 4.** Eight additional qualitative examples on the KappaZeta test set (examples are shown in the same format on the **left** and **right** side of this figure). In each example, columns 1 and 4 are the same as in Figure 3, while column 2 shows pixel-level cloud-type predictions based on thresholding the COTs of an MLP ensemble approach that was trained only on our synthetic data. Column 3 is the same as column 2, but the MLPs were refined on KappaZeta training data. Fine-tuning sometimes yields better results (e.g., top two rows on the left). In many cases, the results are, however, very similar before and after fine-tuning (e.g., third row on the left and right side), and sometimes results worsen after fine-tuning (e.g., fourth row on the left).

### 4.3. Results on Real Data from the Swedish Forest Agency

The Swedish Forest Agency (SFA) is a national authority in charge of forest-related issues in Sweden. Their main function is to promote management of Sweden's forests, enabling the objectives of forest policies to be attained. Among other things, the SFA runs change detection algorithms on satellite imagery, e.g., to detect if forest areas have been felled. For this, they rely on ESA's scene classification layer (SCL), which also includes a cloud probability product. The SFA's analyses require cloud-free images, but the SCL layer is not always accurate enough. Therefore, we applied models that were trained on our synthetic dataset on the SFA's data in order to classify a set of images as 'cloudy' or 'clear'.

To achieve this, the SFA provided 432 Sentinel-2 Level 2A images of size $20 \times 20$ (corresponds to $200 \times 200 \, \text{m}^2$) that they had labeled as cloudy or clear (120 cloudy, 312 clear), where an image was labeled as clear if no pixel was deemed to be cloudy. We note that the cirrus (B10) band was not included, so when working with this dataset, we re-trained our MLP models after excluding this band from the synthetic dataset. Figure 5 shows the locations of these images within Sweden. The 432 images were randomly split into a training, validation, and test split, such that the respective splits have the same ratio between cloudy/clear images (i.e., roughly 28% cloudy and 72% clear images per split). The training, validation, and test sets contain 260, 72 and 100 images, respectively.

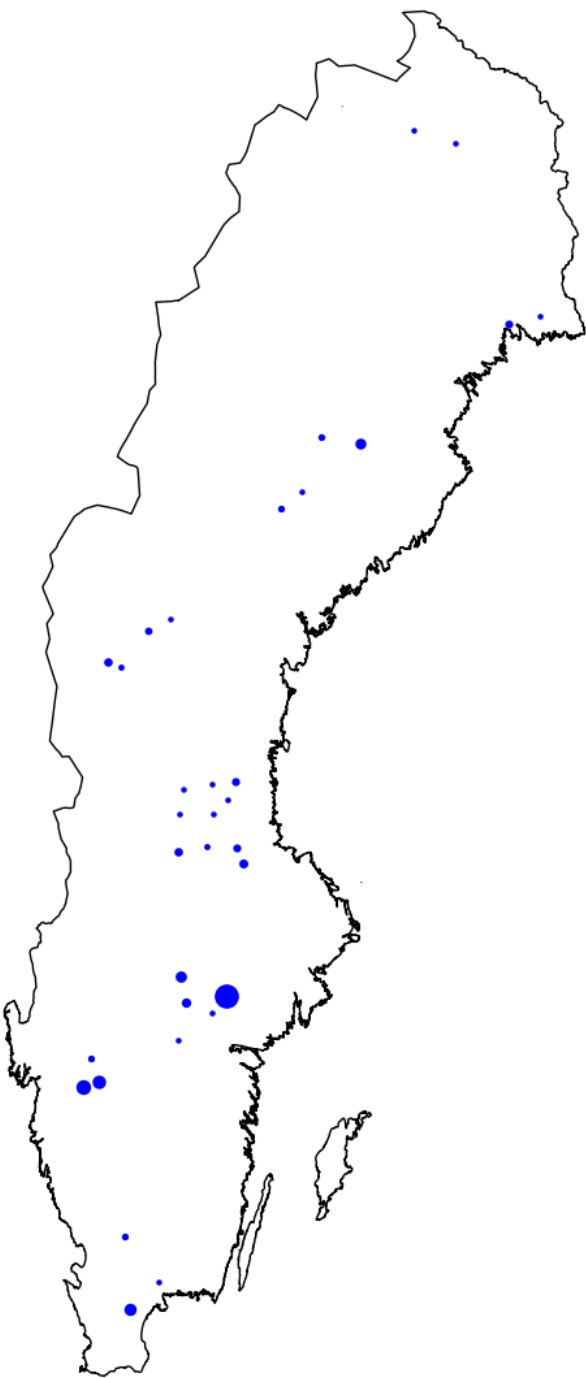

**Figure 5.** Locations in Sweden of the annotated imagery provided by the SFA. Larger dots indicate a higher density of images in the associated region.

The results of the test set are shown in Table 4, which also includes the results of using the ESA-SCL. For our MLPs, we use the validation data in order to set a COT threshold above which a pixel is predicted as cloudy (we found 0.5 to be the best threshold). For the SCL, a pixel is predicted to be cloudy if the SCL label is 'cloud medium probability', 'cloud high probability', or 'thin cirrus'. For both our MLPs and the SCL approach, if a pixel is predicted as cloudy, the overall image is predicted as cloudy. We also compare it with a ResNet-18 model [19] for binary image classification (cloudy and clear, respectively), trained on the union of the training and validation sets. Left–right and bottom–up flipping were used for data augmentation.

**Table 4.** Results on test data from the Swedish Forest Agency (SFA). It is worth noticing that the various MLP approaches were not trained on a single SFA data point. Our MLP-5-ens-10 model yields results comparable to ResNet18-cls (which was trained on the SFA training data), and it outperforms the SCL by ESA. Note that MLP-5 represents the average result among ten MLP-5 models that were trained using different network parameter initializations, whereas MLP-5-ens-10 is an ensemble of those ten different models.

| Metric | MLP-5 | MLP-5-ens-10 | ResNet18-cls | ESA-SCL |
|:---:|:---:|:---:|:---:|:---:|
| F1-avg | 0.73 | 0.88 | 0.90 | 0.68 |
| F1-clear | 0.81 | 0.94 | 0.94 | 0.88 |
| F1-cloudy | 0.65 | 0.82 | 0.86 | 0.48 |
| Rec-avg | 0.73 | 0.86 | 0.91 | 0.66 |
| Rec-clear | 0.77 | 0.97 | 0.94 | 1.00 |
| Rec-cloudy | 0.68 | 0.75 | 0.88 | 0.32 |
| Prec-avg | 0.74 | 0.91 | 0.90 | 0.90 |
| Prec-clear | 0.85 | 0.91 | 0.95 | 0.79 |
| Prec-cloudy | 0.63 | 0.91 | 0.85 | 1.00 |

From Table 4, we see that our main model (an ensemble of ten five-layer MLPs) is on par with the dedicated classification model, despite not being trained on a single SFA data point (except for COT threshold tuning), and despite the fact that the training data represents top-of-atmosphere data (i.e., Level 1C, not Level 2A as the SFA data). We also see that model ensembling is crucial (MLP-5-ens-10 vs. MLP-5), and that our MLP-5-ens-10 model significantly outperforms the SCL (average F1-score 0.88 vs. 0.68 for the SCL). Finally, we note that even using only a single MLP-5 model yields a higher average F1-score than the SCL.

## 5. Conclusions

In this work, we have introduced a novel synthetic IPA dataset that can be used to train models for predicting cloud types of pixels (e.g., clear, semi-transparent, and opaque clouds). In order to enable broad application, in our dataset, we chose to use cloud optical thickness (COT) as a proxy for cloud masking. Several ML approaches were explored for these data, and it was found that ensembles of MLPs perform best. Despite the fact that our proposed synthetic dataset (and thus, associated models) is inherently pixel-independent, the models show promising results on real satellite imagery. In particular, our MLP approach trained on pixel-independent synthetic data can seamlessly transition to real datasets without requiring additional training. To showcase this, we directly applied (without further training) this MLP approach to the task of cloud classification on a novel real image dataset and achieved an F1 score that is on par with a widely used deep learning model that was explicitly trained on these data. Furthermore, our approach is superior to the ESA scene classification layer at classifying satellite imagery as clear or cloudy, and can flexibly generate cloud type segmentation masks via COT thresholding. The code, models, and our proposed datasets have been made publicly available (as of 22 November 2023) at https://github.com/aleksispi/ml-cloud-opt-thick.

**Author Contributions:** Conceptualization, A.P. (Aleksis Pirinen), N.A., R.S. and G.K.; methodology, A.P. (Aleksis Pirinen), N.A., R.S. and G.K.; software, A.P. (Aleksis Pirinen), N.A.P. and R.S.; validation, A.P. (Aleksis Pirinen); formal analysis, A.P. (Aleksis Pirinen), N.A., N.A.P., T.O.T., R.S., C.C. and G.K.; investigation, A.P. (Aleksis Pirinen), N.A., N.A.P., T.O.T., R.S., C.C. and G.K.; resources, R.S., A.P. (Anders Persson), C.C. and G.K.; data curation, R.S., A.P. (Anders Persson), N.A.P. and T.O.T.; writing—original draft preparation, A.P. (Aleksis Pirinen), N.A., N.A.P., T.O.T., R.S., C.C., G.K. and A.P. (Anders Persson); writing—review and editing, A.P. (Aleksis Pirinen), N.A., N.A.P., T.O.T., R.S.,

C.C., G.K. and A.P. (Anders Persson); resources, R.S., C.C. and G.K.; visualization, A.P. (Aleksis Pirinen), N.A.P. and T.O.T.; supervision, A.P. (Anders Persson) and G.K.; project administration, C.C.; funding acquisition, A.P. (Anders Persson), G.K. and C.C. All authors have read and agreed to the published version of the manuscript.

**Funding:** This research was funded by VINNOVA grant number 2021-03643. The APC was funded by VINNOVA grant number 2023-02787.

**Data Availability Statement:** Both the synthetic dataset (described in Section 2), as well as the real data from the Swedish Forest Agency (see Section 4.3), have been made publicly available at https://github.com/aleksispi/ml-cloud-opt-thick (accessed on 5 April 2023). We also conducted experiments on the publicly available KappaZeta dataset (see Section 4.2), and this data was obtained from the following link on 5 April 2023: https://zenodo.org/records/7100327.

**Acknowledgments:** We thank the Swedish Forest Agency for providing the real dataset used in this work.

**Conflicts of Interest:** The authors declare no conflict of interest. The funders had no role in the design of the study; in the collection, analyses, or interpretation of data; in the writing of the manuscript; or in the decision to publish the results.

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
