# Peer review of "Creating and Leveraging a Synthetic Dataset of Cloud Optical Thickness Measures for Cloud Detection in MSI"

_remotesensing, doi:10.3390/rs16040694_

Round 1

Reviewer 1 Report

Comments and Suggestions for Authors

According to the statements in this manuscript, the present work tries to generate a dataset that helps estimate COT given multispectral satellite observations. The technical details about ML models and the relevant experiments are very likely reliable, since they are just common routines including training, validating, testing, and score-based comparison among various alternative models. The goal of these seems to relate the satellite observations of 12-spectral radiance to the COT that is desired, which is considered to be as an application of COT estimation. Besides these, the remainder shown in this manuscript are just those radiative transfer (RT) simulation works, which generate a special dataset and was called as “synthetic COT dataset”. If all above constitute the present study, I do not think that this manuscript is worth to be published, because there is nearly no new point. It seems to be a technical note in which the scientific contribution to the community is absent.

Major
1. The title of this manuscript emphasizes the dataset. But such a dataset generated by RT simulations are just common. It could be derived routinely by simulations of a certain or many RT models, besides the one used here, i.e., RTTOV. In fact, such a table-like dataset that relates the cloud parameters and multispectral TOA radiances is frequently used in traditional cloud retrieval. Hence such a dataset itself does not deserves a “contribution”.

2. If the key point of this study is not the dataset itself, but rather the COT estimation that makes use of ML. As mentioned above, this is no point that could be seemed as new one. For ML to relate the COT and others in this dataset, it is just natural. As for the specific differences in performance among various models and one model to be the optimal one by the comparison, all these are expected. In particular, even for the COT estimation, the approach proposed shows to be too simple to capture the real problem. For example, cloud type was designated as one input. This is unreasonable in practice. Given a satellite image, the presence/absence of cloud, cloud optical thickness, cloud type, and others, are all definitely unknowns, even though those atmospheric and surface background information were taken to some extent as prior knowns. So the value of such a COT estimation approach is questionable.

3. In Figure 1, there are three cloud categories, why were they introduced herein? Is this necessary? It is pointed out that the sum is not 50,000. It is 200,000? What do these COT distributions about the three cloud categories mean in this context? What do they affect?

4. Here the 12 spectral bands are from the MSI onboard the Sentinel-2A. It is not important that what these specific wavelengths are, although they are supposed to be common channels in visible and infrared band. However, these 12-band based dataset certainly has no contribution to others that do not duplicate the MSI. Furthermore, there is no difficulty to simulate such a dataset for the spectral band configuration of other satellite platforms. So the dataset itself is not a general one and has no practical value. In other words, very few people can benefit from this dataset. In fact they do not need this dataset and can readily create their own.

5. Anyhow, I cannot find anything in this manuscript that is valuable. The concrete works are substantial, but they are too plain to be called as a “study” or “research”. They are mostly common results from common treatments.

Comments on the Quality of English Language

It is well.

Reviewer 2 Report

Comments and Suggestions for Authors

General comments

This manuscript proposes a simulation-generated synthetic dataset for cloud optical thickness estimation, which helps to solve the problem of few measured COT data in practical applications, and achieves methodological advances for the cloud optical thickness estimation by introducing deep learning methods. The article chooses the data open access method, which enables the research results to be more widely disseminated and shared, and helps to promote academic communication and knowledge dissemination.

Specific comments:

line 104-119: In this paragraph, the setup of cloud attribute and characterization in the radiative transfer model should be elaborated with more detail and clarity, since clouds are the focus of this paper.

How the geometric thickness of the cloud is set in the model RTTOV-v13?

line 117: What does "consistency-check" mean? Please explain in the context.

line 139: What exactly does "ground truth COT yi" mean? Does it mean that the COT calculated by the model is taken as the truth value?

How are the satellite angle and sun angle set in the radiative transfer model? How many groups were selected respectively? When the machine learning model was trained, were these groups trained together? Please add some explanations.

How are the surface types set in the radiative transfer model? Especially for transparent or translucent clouds, their values are important factors for the radiance observation of satellite sensors, i.e., imagers.

It would be good to do some sensitivity analyses, e.g., how much does the angle, surface conditions mentioned above, affect the radiative transfer model results?

The data used for training in this study are all from simulated data, has there been any attempt to include some real observation data, such as Sentinel-2 observation data, which after all have been in operations for years.

Regarding the testing and validation of the COT results, it is recommended that the use of real COT data other than Sentinel-2 will make the results of the test more convincing.

Reviewer 3 Report

Comments and Suggestions for Authors

The authors propose a synthetic dataset for cloud optical thickness (COT) estimation.  Simulations are performed for different cloud types, COTs and ground surface/atmosphere profiles and have been tailored to TOA radiances for 12 spectral bands monitored by the Multi-Spectral Instrument (MSI) onboard Sentinel-2 platforms.  Trained Machine Learning (ML) models are used to predict COT based on the synthetic data and real satellite image datasets.

The paper is well written although I found it to be rather specialized to the machine learning community.  The overall results for COT determinations and for cloud/clear sky determinations appears to be an advancement over current state-of-the-art ML methods.  Therefore, the paper will be of interest to the remote sensing community involved in ML methods.  I recommend publication following consideration of the issues below.

1. It is not clear what the 3D effect is in line 126. This effect should be described.

2. In my view, the nomenclature, R^19 and R_+ is not common knowledge and should be defined.

3. Explain what overriding notation means to let x_i belong to R^12 in line 138.

4. Perhaps those practiced in ML would know how to interpret “real imaginary I belongs to R^HxWxC”.  I feel the product HxWxC should be defined.

5. The paragraphs from 150 to 193 is difficult to interpret and appears to be written for a computer scientist.

Round 2

Reviewer 1 Report

Comments and Suggestions for Authors

Unfortunately, I do not find any meaningful revision on this manuscript, although I have given a few major comments, which are mainly about the lack of this study. To avoid possible misunderstanding, I have carefully read the replies from the authors. Now I have to insist on my original view that there are severe deficiencies in this manuscript. The primary reasons are as follows.

The main work is the generation of a dataset that connects 12 spectral radiances with an optimal estimation of COT. Such a dataset of course has some values for others, especially for those who are not familiar with these works (ready-made, lowering threshold, easy participation, and so on, as mentioned by the authors). But this is not enough for a necessary and satisfied study.

It is apparent that this work contains two parts. One is the RT simulations for generating the basic dataset and the other is the ML modeling for establishing the connection. For the first part, such a dataset is inherently derived from the RT model outputs, with just ordinary configurations. This part just used the most common tool and got consequent outputs. So here there is no contribution from the study itself. Then the second part, as I mentioned previously, it is just common operation and the results are also as expected. It is natural to find an optimal machine learning model in these comparisons. This part is of course meaningful, but is very limited.

Overall, the authors could emphasize the importance of such a dataset (considered to be benchmark), and I admit this dataset could be useful to some others. However, potential readers can get little from this manuscript, since new ideas, new methods or new findings are absent.

Actually, the main contents of the manuscript also support the above judgements. The four Tables are mostly the detailed records about comparisons, helping the author to make selection among MLP methods. As for the five Figures, Figure 1, 2, and 5 are all schematic and provide little information. They are in fact unnecessary in this manuscript. The remained two, Figure 3 and 4, are similar and just qualitative. Given these as the main results, they are so few to support a study making sense. Even for a technical-note type, there is much room for improvement. Anyhow, for a formal study, these results are far from adequate and I think they have not been well designed.

Comments on the Quality of English Language

English language is well.

Author Response

Our reply / input to the second round of comments from Reviewer #1:

We agree with the reviewer that the reflections are calculated by the radiative transfer model and thus by a common tool. However, we still do not agree that this alone makes the generation of such a training dataset trivial. The compilation and preparation of input data (surface reflection, background atmosphere) is not part of the model but crucial for later success in training an artificial neural network (ML model). The same applies to the composition of the dataset (e.g. COT distribution) and the combinations of the parameters (e.g. cloud type/height with background profile). This all affects the training phase and can yield a completely different ML model behavior after it has been trained. Errors in compiling such a dataset do not matter as long as the ML methods are only verified within this dataset. However, these inaccuracies would become apparent when applied to real measured values, long after the dataset was actually created.

So again, as also emphasized in our original reply in the first round of reviews, we are convinced that compiling a synthetic dataset that proved its function (based on extensive experimentation using multiple ML-based approaches) on real data is not trivial and is worth publishing. Finally, we have also included a separate subsection in Section 1, that summarizes the main contributions of this work.

Reviewer 2 Report

Comments and Suggestions for Authors

Thanks.

Author Response

We wish to again thank this reviewer for insightful comments and suggestions. We appreciate that the reviewer took the time to read our previous responses.